# Unleashing the anti-tumor angiogenic potential of nano-formulated orientin: *In Silico, In Vitro*, and *In Ovo* studies

**Yashwanth Elumalai**[1], **Kathiresan Nachammai**[2], **Kirubhanand Chandrasekaran**[3‡],
**Langeswaran Kulanthaivel**[2], **Sharon Benita Stephen**[1‡], **Kanu Shil**[4‡],
**Ram Kumar Anandan**[5], **Abdulhadi Ibrahim Bima**[6], **Zeenath Khan**[7],
**Abdulhadi S. Burzangi**[8], **Noor A. Shaik**[9], **Nuha Al-Rayes**[10,11]*,
**Gowtham Kumar Subbaraj**[1]*

**1** Faculty of Allied Health Science, Chettinad Hospital and Research Institute, Chettinad Academy of Research and Education, Kelambakkam, Tamil Nadu, India, **2** Department of Biotechnology, Alagappa University, Karaikudi, Tamil Nadu, India, **3** Department of Anatomy, AIIMS, Nagpur, India, **4** Faculty of Paramedical Sciences, Assam Down town University, Guwahati, Assam, India, **5** Department of Biotechnology, School of Bioengineering, SRM Institute of Science and Technology, Kattankulathur, Tamil Nadu, India, **6** Department of Clinical Biochemistry, Faculty of Medicine, King Abdulaziz University, Jeddah, Saudi Arabia, **7** Department of Science, Prince Sultan Military College of Health Sciences, Dhahran, Saudi Arabia, **8** Department of Clinical Pharmacology, Faculty of Medicine, King Abdulaziz University, Jeddah, Saudi Arabia., **9** Department of Genetic Medicine, Faculty of Medicine, King Abdulaziz University, Jeddah, Saudi Arabia, **10** Medical Laboratory Sciences, Faculty of Applied Medical Sciences, King Addulaziz University, Jeddah, Saudi Arabia, **11** Princess Al-Jawhara Al-Brahim Centre of Excellence in Research of Hereditary Disorders, King Abdulaziz University, Jeddah, Saudi Arabia

‡ These authors share first authorship on this work.
* gowtham_phd@yahoo.com (GKS), nalrayes@kau.edu.sa (NA)

## Abstract

The study investigated the potential of nano-formulated orientin (NF-O) in the anti-angiogenic cancer therapy. Orientin is a flavonoid that has a promising effect against anti-inflammatory, anti-oxidant, and anti-arrhythmia properties. Nano-formulation aimed to overcome this limitation and also served to enhance its therapeutic efficacy. In silico docking studies, the favorable binding of orientin was identified with the key oncogenic targets (EGFR, ALK, KRAS, NTRK). After nano-formulation, UV spectroscopy confirmed the integrity of orientin with no shift in the λmax (347 nm). Dynamic light scattering showed a significant reduction in the improved particle size (PDI decreased from 0.863 to 0.173) by nano-formulation from 559 nm to 220 nm. Fourier Transform infrared spectroscopy analysis confirmed that the nano-formulation process did not alter the chemical structure of orientin. In-vitro studies using MCF-7 breast cancer cells showed that NF-O inhibited cell growth and reduced viability in a dose-dependent manner. At 10 μM, NF-O significantly inhibited the cell growth and migration compared to the control and native orientin in wound healing assays ($p < 0.01$). In ova, using the chick chorioallantoic membrane (CAM) assay, NF-O (10 μg/ml) significantly inhibited angiogenesis by reducing blood vessel density, branching, length, and network formation compared to controls and native orientin. These

**Data availability statement:** All relevant data are within the paper and its Supporting Information files.

**Funding:** This research work was funded by Institutional Fund Projects under grant no. (IFPIP: 1174-290-1443). The authors gratefully acknowledge technical and financial support provided by the Ministry of Education and King Abdulaziz University, DSR, Jeddah, Saudi Arabia The funders had a pivotal role in data collection, experimental design, and analysis of the manuscript.

**Competing interests:** The authors have declared that no competing interests exist.

**Abbreviation:** NF-O, Nano-formulated Orientin; CAM, Chorioallantoic Membrane; DLS, Dynamic light scattering; FT-IR, Fourier transform infrared spectroscopy; DMEM, Dulbecco's modified eagle medium; FBS, Fetal Bovine serum; DMSO, Dimethyl sulfoxide; PBS, Phosphate buffer saline; PDI, Polydispersity index value; VEGF, Vascular endothelial Growth factor; FGF2, Fibroblast growth factor 2; DOX, Doxorubicin.

findings suggest that NF-O holds significant promise as a novel anti-angiogenic agent for the cancer treatment.

---

## 1. Introduction

Cancer is a major global cause of mortality, marked by uncontrollable cell division and cell growth. This disease is distinguished by various traits, such as uncontrolled cell proliferation, sustained angiogenesis, and insensitivity to growth signals, invasive behavior, and resistance to apoptosis [1]. It is a major public health issue on a global scale and stands as the top second reason for mortality in the United States. Among females, Breast cancer is the second most prevalent kind of malignancy and is frequently diagnosed. The literature provides evidence that prior and ongoing research has a substantial influence on improving the clinical outlook for breast cancer [2]. According to the National Cancer Registry Programme (NCRP), India, the incidence of cancer cases is likely to increase from 1.46 million in 2022 to 1.57 million in 2025 [3]. Angiogenesis, a vital process in cancer growth, is essential for solid tumors to thrive as they need a blood supply. Tumors stimulate new blood vessel growth (angiogenesis) to supply themselves with nutrients and oxygen, fuelling growth, invasion, and ultimately, the spread of cancer cells (metastasis) [4]. Several factors, including VEGF, FGF-2, PDGF, and others, promote angiogenesis in tumors. VEGF, MMPs, and FGF2 are key drivers, while angiopoietins and interferons can inhibit the process [5]. Flavonoids, a group of polyphenolic compounds, are abundantly present in various vascular plants. These phytonutrients can be commonly found in vegetables, fruits and Herbal plants. Flavonoids come in diverse forms, including free forms, glycosides, and methylated derivatives [6]. Flavonoids have a long history of usage in medicine as, neuroprotective, anti-cancer, anti-bacterial, anti-malarial anti-viral, antioxidant, anti-proliferative and anti-angiogenic agents [7]. Flavonoids combat cancer at all stages, including metastasis. They also target key regulators of cancer spread, such as those involved in cell transitions, and molecules like MMPs, uPA/uPAR, and TGF-β [8]. Orientin, formerly known as luteolin-8-C-glucoside, is a flavonoid compound that is glycosylated and easily soluble in water that is isolated from a variety of plants, consisting of Ocimum sanctum, various Phyllostachys species, different Passiflora species, and various Trollius species. It is mostly found in rooibos tea [9]. Extensive investigation into the medicinal properties of orientin has revealed its potential as a valuable therapeutic agent. It exerts diverse pharmacological effects including anti-inflammatory, antioxidant, cardioprotective, antitumor, and neuroprotective properties [10]. Despite these encouraging findings, the specific processes underlying these therapeutic effects have not been completely investigated and remain unknown. Furthermore, it is crucial to acknowledge that orientin's hydrophilic nature poses obstacles in terms of traveling the blood-brain barrier [11]. According to Sharma et al., It has been discovered that orientin significantly suppresses the development of human liver cancer cell lines by efficiently inhibiting their proliferation [12]. Orientin demonstrates the ability to inhibit proliferation in esophageal cancer and induce apoptosis, a programmed cell death

process [13]. Additionally, it has been observed to regulate the COX-2/PGE-2 pathway in lung cancer cell lines [14]. Furthermore, Orientin has been noted for its ability to inhibit nuclear factor-kappa B transcription in colorectal cancer and human bladder T24 cancer cells, indicating a potential mechanism behind its anticancer effects [15]. In addition, orientin has been shown to inhibit tumor invasion and cell migration, suggesting its potential as a therapeutic agent targeting cancer metastasis [16]. Research literature has shown that Orientin exhibits lower solubility in both aqueous and organic solvents, such as ethanol, dimethyl sulfoxide, and acetone, which is similar to that of other bioactive chemicals [17]. Through size distribution reduction and surface modification, nanotechnology has shown promising results in enhancing the soluble content, bioavailability, and biological activity of flavonoid compounds. [18]. Recently, the chorio-allantoic membrane (CAM) assay has become a highly effective tumour model. It provides a reliable and suitable system for assessing the ability of nanoparticles to deliver anticancer drugs [19]. The CAM assay is preferred over other in vitro or in vivo models because of its high sensitivity and cost-effectiveness. The novelty of the study includes the latest advancement of nanotechnology has the potential to transform drug delivery and greatly enhance the solubility of poorly soluble drugs. Due to their increased surface area and ability to penetrate biological barriers, nanoparticles are highly effective for drug delivery. These innovations could lead to the creation of more efficient and potent drug formulations for treating various diseases, including cancer. We aimed to assess the potential ability of native and nano-formulated orientin against the cancer microenvironment. The parameters such as molecular docking, nano-characterization, Cell culture analysis, and the *In ova* CAM assay were performed.

## 2. Materials and method

### 2.1. Ethical approval

Chick embryos younger than embryonic day 13 (E13) are assumed unable to experience pain. Therefore, no ethical permission is needed as per the regulations of the country. No ethical clearance is needed.

### 2.2. Materials

**2.2.1. Chemicals.** Orientin with a purity level of up to 98% were obtained from TCI Chemicals Company, Japan. Chitosan was purchased from Himedia Laboratories, India. Hexane and Sodium Dodecyl Sulfate (SDS) were obtained from SRL Chemicals, India. Sodium tripolyphosphate was purchased from LOBA Chemicals, India. The MCF-7 cell lines (human breast cancer) were obtained from NCCS, Pune, India. Dulbecco's Modified Eagle Medium (DMEM), Fetal Bovine Serum (FBS), 3-[4, 5-dimethylthiazol-2-yl]-2, 5- diphenyltetrazolium bromide (MTT) and 4′, 6-diamidino-2-phenylindole (DAPI) were obtained from Himedia Laboratories, India. Fertilized chicken eggs were obtained from Tamil Nadu Veterinary and Animal Sciences University, Tamil Nadu, India. All the chemicals, including solvents used in the study, were of analytical or laboratory-grade quality, ensuring compliance with standard protocols.

**2.2.2. Cell culture maintenance.** For cell growth and proliferation, the MCF-7 cells were procured from NCCS, Pune, INDIA. The MCF-7 breast cancer cell lines were cultured using DMEM media consisting of 10% FBS and 1% penicillin-antimycotic solution. The cell culture flask was maintained in a 5% $CO_2$ incubator at 37°C for further analytical modules.

## 3. *In silico* analysis

### 3.1. Target and lead molecule preparation

The RCSB Protein Data Bank was used to retrieve the target receptor molecules such as EGFR (PDB ID: 2ITY), KRAS (PDB ID: 7LGI), NTRK (PDB ID: 7VKO), and ALK (PDB ID: 2XP2). Initially, protein structures undergo preprocessing processes that remove all water molecules and heteroatoms. The protein receptors were then stored as PDBQT files in the PyRx workspace folders, missing hydrogen and charges were added. The Orientin SDF file was accessed using PyRx integrated Open Babel module. These files were then energy minimized using the MMFF94 force field and the steepest

descent algorithm with 2000 steps. Finally, in PyRx v0.8, Open Babel was used to transform all energy-minimized structures to PDBQT format [20].

## 3.2. Molecular docking simulation

The orientin compound and protein structures were imported into PyRx, the virtual examining software. The conjugate gradient method and the Universal Force Field (UFF) were used to conduct energy reduction. Following that, PyRx's integrated Open Babel tool was used to store both chemical compounds and protein structures in '.pdbqt' format. The dimensions and coordinates of the grid box were modified by either tracing the boundaries or entering precise values into the corresponding fields. PyRx uses the Lamarckian genetic algorithm for conformational search [20]. The docked protein-ligand complexes were subjected to further analysis using Discovery Studio 2021 to have a greater understanding of their 2D & 3D interactions [21,22].

## 4. Synthesis and characterization of Nano-formulated Orientin by using solvent evaporation method

The orientin drug was transformed into a nano-formulation using the solvent evaporation method. Initially, 10 mg of Orientin drug was precisely measured and dissolved in 2 ml of ethanol. The mixture was meticulously blended, and then 4 mg of SDS was added and mixed well. All the mixtures were transferred into a beaker, which was subsequently placed on a magnetic stirrer set at a speed of 200 RPM. During stirring, Hexane was added drop by drop. The beaker was left undisturbed for 4 hours to allow the solvent to evaporate. Once the solvent gets evaporated, the sample was collected and placed in a dark room overnight. The following morning, the sample was collected and kept at 4°C for later usage as shown in Fig 1.The physiochemical properties of the synthesized Nano- particles were evaluated using Fourier-transform infrared spectroscopy (FT-IR), UV spectrophotometry and Dynamic light scattering (DLS).

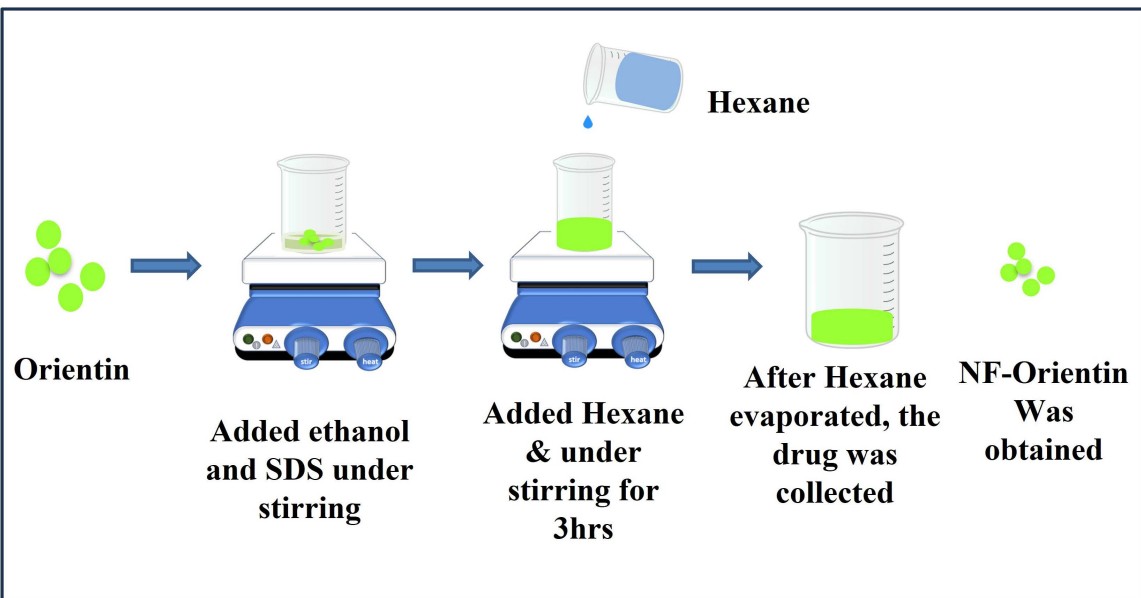

**Fig 1. Synthesis preparation of nano-formulated orientin by using solvent evaporation method.**

## 5. Characterization of nanoparticles

To examine the functionalization of nanoparticles, all experimental groups of nanoparticles underwent several characterization techniques. The shift in the peak of the native, NF-O was analyzed using UV-visible spectrophotometry. The size of the nanoparticles was determined through DLS techniques. Further, the samples functional groups were identified and evaluated using FT-IR spectroscopy.

### 5.1. UV-spectrophotometer

The UV-Spectrophotometer was utilized to ascertain the maximum absorbance of native, nano-formulated orientin. A solution containing 1 mg/ml of the drug in water was prepared and placed in a cuvette for measurement, and the resulting spectrum was recorded. Similarly, the spectrum of the functionalized native, NF-O drug was also analyzed using the UV spectrophotometer. All these recordings were performed using a UV-1800/Shimadzu UV-Spectrophotometer.

### 5.2. Dynamic light scattering

By measuring the hydrodynamic diameter (DH) of the particles, Dynamic Light Scattering (DLS) is a frequently used technique to analyze the size distribution of particles in a liquid media. In this specific technique, 50µl of the prepared nanoparticles is mixed with 950µl of Milli-Q water, and the samples are sonicated for 20 minutes to achieve a uniform dispersion. Subsequently, the resulting suspension is transferred to a cuvette for measurement using DLS. For the present study, the instrument used for these measurements was the nano-zs 90 Zeta sizer from Malvern.

### 5.3. Fourier transform infrared spectroscopy

FT-IR spectroscopy serves as a primary method for identifying the functional groups present in materials. Each peak observed in the spectrum corresponds to a specific functional group. In this study, a small quantity of native, NF-O drugs was placed, and the peaks were recorded. The samples underwent analysis using Bruker alpha instruments in FT-IR mode, and the resulting spectra were recorded.

## 6. Cell viability assay

To assess the viability of MCF-7 cells, the MTT assay was employed. A suspension containing 10,000 cells per well was prepared and allow to develop for around 24 hours in a 96-well plate. Once the cells reached the desired growth stage, they were treated with appropriate quantities of native orientin and NF-O. To assess the efficacy of the therapy, untreated cells were utilised as the negative control and DOX (5µM) as the positive control. Following this, the plates were placed in an incubator for 24 hours at 37 degrees Celsius with a slightly increased level of carbon dioxide (5%), with close monitoring throughout the process. After removing the MTT solution, the cells were exposed to DMSO. Then, the absorbance was measured at a wavelength of 570 nm using a microplate reader.

### 6.1. Cell migration assay

MCF-7 cell migration was assessed by culturing them in 6-well plates at a density of 8 x 10^4 cells/well with complete DMEM media. After removing the PBS and serum-free DMEM media, the MCF-7 cells were washed to form a monolayer with approximately 75% confluence. A scratch wound was created using a sterile 200µl tip, and any floating or suspended cell debris was removed by rinsing the wells with a serum-free medium. Subsequently, orientin therapy was administered to the MCF-7 cells at a various concentration. And they were incubated for up to 24 hours. An inverted microscope was utilized to capture images of the scratch wound at different time interval. Using the ImageJ application version 1.8, the wound area's size in each picture was measured to track the migration of cells into the gap area. Images of the scratch gap distance were taken under the microscope at a magnification of 20X [23].

## 7. *In ova* CAM assay

### 7.1. Anti-angiogenesis activity

To commence the experiment, fertilized chicken eggs were acquired and placed inside an incubator programmed to maintain at 37°C with the humidity level of approximately 53%. On the first day, the eggshells were cleaned with ethanol, and a small window (2x2cm) was created by carefully drilling into the eggshell. Using a sterile gauge needle, 3 ml of albumin content was removed. On the third day, under a controlled environment, a sterilized disc was gently positioned on the fully-grown CAM. To the sterile disc, native orientin, NF-O at a concentration of 10 µg/ml were loaded. The Fig 2 illustrates the stepwise procedure of in ovo experimentation, including egg incubation, puncture, treatment administration, and sample collection at specific developmental stages. The present study was categorized into 4 groups: Group I act as control and did not receive any treatment with drugs. Group II act as native orientin treated, Group III act as NF-O treated. Following the treatment, the eggshells were carefully sealed using sterile tape. After incubating the eggs for five days at 37°C and 53% humidity, researchers photographed the membranes (CAM) through openings made in the eggshells. These membranes were then preserved with formalin and stored at −20°C for further analysis.

### 7.2. Histopathological analysis

The CAM treated with the drug were immersed in a 10% Formaldehyde fixative solution. Subsequently, using forceps, a 1 cm$^2$ section of the surrounding membrane area which is treated was rigorously removed. This section was then dehydrated using various alcohol grades and embedded using paraffin wax. Using a rotating microtome (Weswicox, Japan), vertical tissue slices with a thickness of 6µm were obtained. The tissue slices were subjected to a gradual process of alcohol treatment, beginning with absolute alcohol and followed by 50%, 70%, and 90% concentrations. After cleaning with xylene, these tissue slices were coloured with hematoxylin and eosin stain. For qualitative assessment, a light microscope

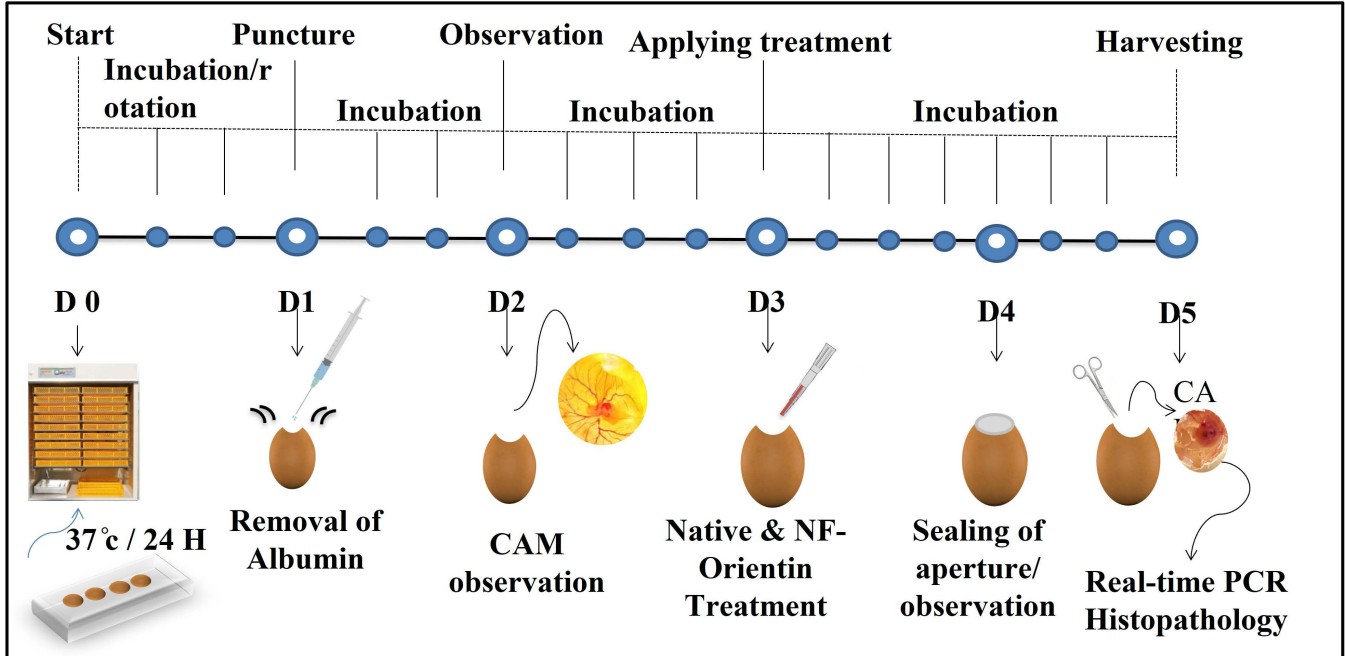

**Fig 2. Illustrates the schematic representation of *in ova* experimentation showing with timeline.**

at 40X magnification was used to examine the mounted tissue sections. Snapshots were taken using a light microscope featuring Nikon camera, providing a 10X magnified view.

### 7.3. Gene expression analysis using RT-PCR

After isolating RNA that had been converted into cDNA using the Takara kit technique, the total amount of RNA from CAM membrane was extracted using the TRIzol method. Table 1 lists all of the primer sequences that were employed in the current investigation. β-actin was regarded as a house keeping gene. SYBR green master mix was used in the RT-PCR test to amplify the interest gene. Initial denaturation at 90° C for 30 s, annealing at 60°C for 30s, and extension at 72°C for 30s for 40 cycles were programmed during amplification.

### 7.4. Statistical analysis

The results are presented as mean ± SD based on the number of experiments conducted. Statistical analysis was performed using one-way ANOVA in GraphPad Prism 8. Following a significant ANOVA result, the Newman-Keuls post-hoc test was applied to compare group means. A significance threshold of $p < 0.05$ was set to determine statistically significant differences between groups.

## 8. Results

### 8.1. Molecular docking

Molecular docking plays a crucial role in drug designing by assessing the binding mechanism and efficacy of active compounds [24]. Molecular docking was conducted and the outcomes are recorded in Table 2.

### 8.2. Binding poses of EGFR with orientin

The active lead molecule binds to the EGFR receptor (identified by PDB ID 2ITY) through various interactions including hydrogen bonds, pi-sigma bonds, and pi-alkyl bonds. Four conventional hydrogen bond interactions were established between the orientin and EGFR residues such as MET 793 LYS 745, ASP 855, and ARG 841. Moreover, Pi-sigma interaction took place with the residues LEU 844 and LEU 718. Additionally, carbon hydrogen bond interaction was formed by ASP 855. Furthermore, four Pi-Alkyl interactions were observed with the EGFR residues such as VAL 726 and ALA 743. 2ITY_orientin demonstrated significant interactions, with a docking score of −8.4 kcal/mol, two and three dimensional are displayed in Fig 3A & 3B.

### 8.3. Binding poses of ALK with orientin

The protein ALK (PDB ID: 2XP2) engages in many forms of interactions with orientin, including carbon hydrogen bonds, conventional hydrogen bonds, pi-sulfur interactions and pi-anion contacts. The compound 2XP2_orientin had a docking score of −6.4 Kcal/Mol. This score was supported by a visual representation of its two and three-dimensional interactions, which can be seen in Fig 3C and 3D. The docking result indicate that the ALK protein residues such as GLN 1146, LEU 1198, GLU 1197, and GLN 1177 are involved in formation of four Conventional hydrogen bonds with orientin. Carbon hydrogen bond and Pi-Anion interaction was formed by GLU 1197. When a cysteine 1259 residue exhibits a pi-sulfur interaction, it usually indicates that an aromatic ring is engaging with a sulfur atom.

**Table 1. Primer Sequence expression for VEGF-A and FGF2.**

| Mutant | Forward Primer (5') | Reverse Primer (5') |
|---|---|---|
| VEGF-A | CAATTGAGA CCCTGGTGG AC | GCTTGCTGT GCTCTTAGC C |
| FGF2 | ATGGCTGCCCAAGCTGC | TCAGCTTCCTTGTAGCCTTC |

**Table 2. Interaction profile of targeted receptor molecules and orientin.**

| S.No. | Protein (PDB ID) | Docking score (kcal/mol) | Interaction | Interacting Residues | Bond Distance (Å) |
|---|---|---|---|---|---|
| 1. | EGFR (2ITY) | −8.4 | Conventional Hydrogen bond | MET 793 | 5.46 |
| | | | | LYS 745 | 4.78 |
| | | | | ASP 855 | 3.28 |
| | | | | ARG 841 | 5.69 |
| | | | Pi-Sigma | LEU 844 | 5.47 |
| | | | | LEU 718 | 5.44 |
| | | | Pi-Alkyl | VAL 726 | 5.47; 5.08; 5.53 |
| | | | | ALA 743 | 6.15 |
| | | | Carbon Hydrogen Bond | ASP 855 | 5.69 |
| 2. | ALK (2XP2) | −6.4 | Conventional Hydrogen bond | GLN 1146 | 4.05 |
| | | | | LEU 1198 | 5.16 |
| | | | | GLU 1197 | 6.10 |
| | | | | GLN 1177 | 3.79 |
| | | | Carbon Hydrogen Bond | GLU 1197 | 4.12 |
| | | | Pi-Anion | GLU 1197 | 4.66 |
| | | | Pi-Sulfur | CYS 1259 | 6.90 |
| 3. | KRAS (7LGI) | −8.24 | Conventional Hydrogen Bond | TYR 32 | 4.62;5.72 |
| | | | | ASN 116 | 5.06 |
| | | | | ASP 30 | 4.61 |
| | | | Carbon Hydrogen Bond | GLY 15 | 3.70 |
| | | | Pi-Pi T-shaped | PHE 28 | 6.64;6.37 |
| | | | Pi-Alkyl | ALA 18 | 6.76 |
| | | | | ALA 146 | 6.84;6.71 |
| | | | | LYS 117 | 5.78;5.20 |
| | | | | LEU 120 | 5.94 |
| | | | | LYS 147 | 5.06 |
| 4. | NTRK (7VKO) | −8.3 | Conventional Hydrogen Bond | ASP 668 | 3.32 |
| | | | | GLY 667 | 4.96 |
| | | | | ARG 673 | 3.58;5.30 |
| | | | | ASP 596 | 3.28;3.61 |
| | | | | MET 592 | 4.62 |
| | | | Pi-Sigma | LEU 657 | 5.53;5.90 |
| | | | Pi-Alkyl | VAL 524 | 5.07;5.07 |
| | | | | ALA 542 | 6.71 |
| | | | | LEU 516 | 5.27;4.47 |

## 8.4. Binding poses of KRAS with orientin

KRAS (PDB ID: 7LGI) interacted with orientin forms different interactions such as conventional hydrogen bond, carbon hydrogen bond, pi-alkyl and pi-pi t-shaped. 7LGI_ orientin has a docking score of −8.24 Kcal/Mol and its extensive two and three-dimensional interactions are shown in Fig 4A & 4B. Four conventional hydrogen bond interactions formed by KRAS protein residues such as TYR 32, ASN 116, and ASP 30. A carbon-hydrogen bond is generated by the residues GLY 15 serve critical functions in maintaining protein structure and function. Two Pi-Pi T-shaped interactions were formed by PHE 28. Five Pi-Alkyl interactions were formed by ALA 18, ALA 146, LYS 117, LEU 120, and LYS 147.

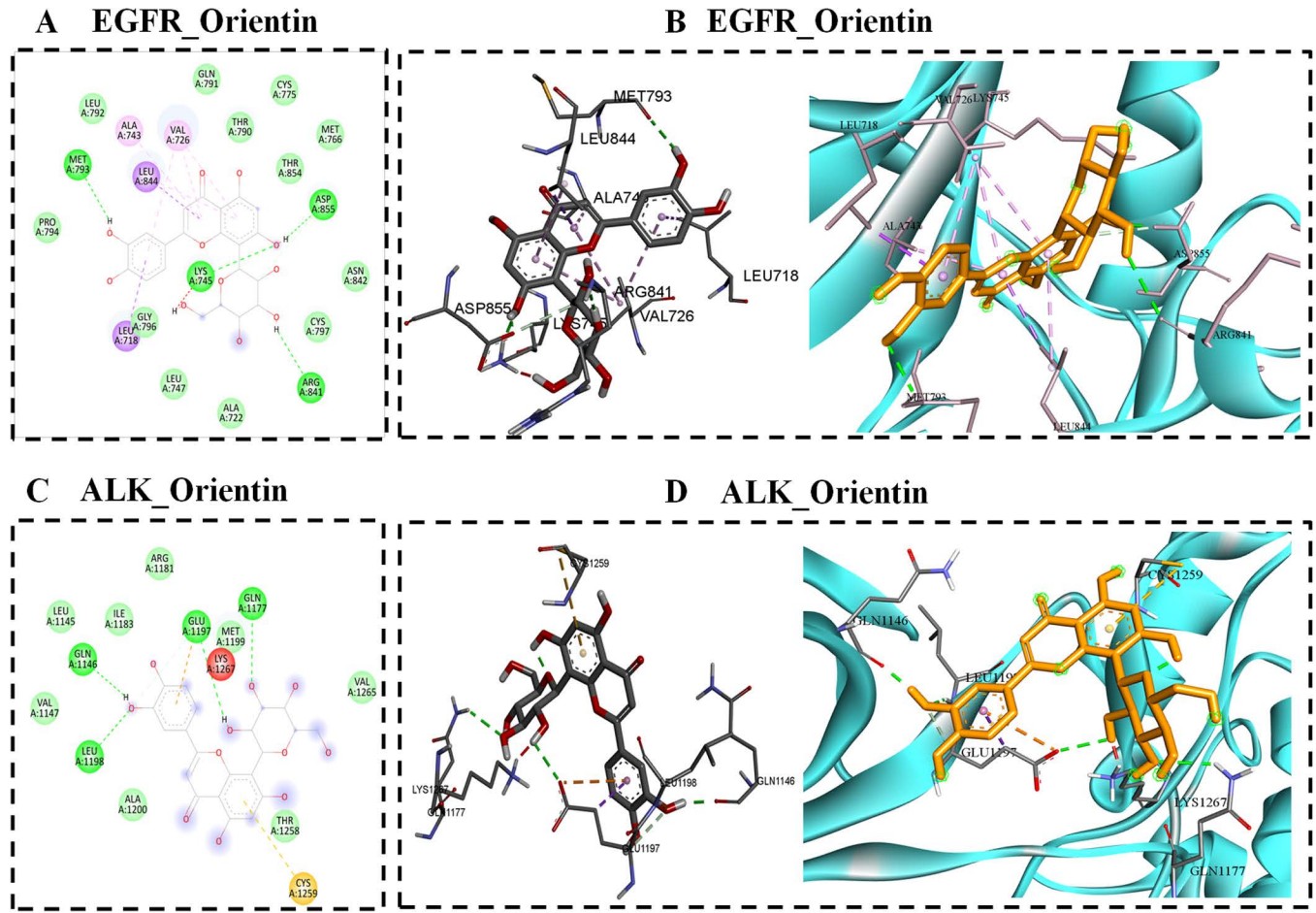

**Fig 3.** A) EGFR-orientin docked complex represented in the 2-d interaction, B) 3-d interaction of EGFR-orientin, C) 2-d interaction between ALK and orientin, and D) 3-d interaction patterns between ALK and orientin.

### 8.5. Binding poses of NTRK with orientin

The docking score between NTRK and orientin −8.3 Kcal/Mol, as seen in Fig 4C & 4D, which shows the complex's two and three-dimensional interactions. Seven conventional hydrogen bonds are formed by ASP 668, GLY 667, ARG 673, ASP 596, and MET 592. Two Pi-Sigma connections are established by the Leucine amino acid residue located at position 657. Additionally, five alkyl interactions are formed by residues such as VAL 524, ALA 542, and LEU 516. *In silico* docking simulations with angiogenesis-related targets EGFR (PDB ID: 2ITY), KRAS (PDB ID: 7LGI), NTRK (PDB ID: 7VKO), and ALK (PDB ID: 2XP2) show high affinity for molecular targets that regulate angiogenic signaling. This is validated by our *in ova* CAM assay where the density of the vascular network was highly decreased on treatment with NF-O.

### 9. Characterization of native orientin and NF-orientin using UV-spectrophotometer

A UV-Spectrophotometer was employed to detect the presence of Native, NF, NE-Orientin, and the resulting spectra ranged between 200 and 800nm. The λ max of native Orientin was found to be 347 nm (A), indicating its maximum absorption wavelength. Similarly, the λ max of the NF-O formulation was also observed to be 347 nm (B). Subsequently,

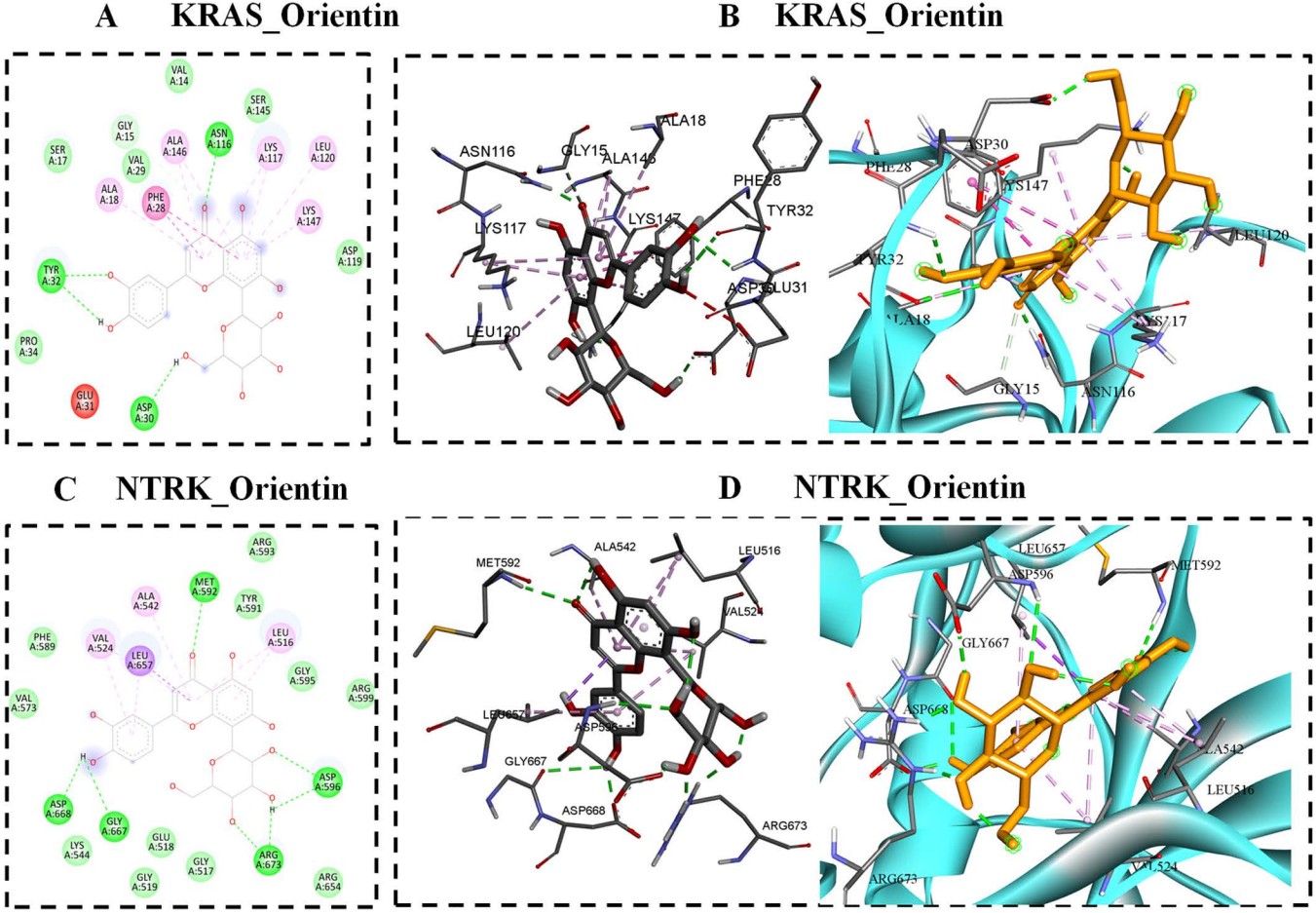

**Fig 4. A) KRAS-orientin docked complex illustrated in the 2-d interaction, B) 3-d interaction of KRAS-orientin, C) 2-d interaction between NTRK and orientin, and D) 3-d interaction patterns between NTRK and orientin.**

a combined spectrum (C) was generated, displaying the absorption characteristics of both native Orientin and the NF-O formulation as shown in Fig 5.

### 9.1. Characterization of orientin and NF-O using DLS

Dynamic light scattering (DLS) analyses was performed for comparing the dimensions of the freshly synthesised NF-O formulations with the initial structure of orientin. To achieve proper dispersion, the NF-O underwent sonication for one hour before the DLS analysis. The control orientin was analysed for its particle size distribution, which stayed to be 559 nm, of PDI value of 0.863. In contrast, the distributed size of NF-O was observed to be 220 nm, of PDI value of 0.173. These findings indicate a significant reduction in particle size and a decrease in the polydispersity index to below 0.25 with the nano-formulation of the drug in order to over the solubility issue as shown in Fig 6. Overall, the results demonstrate that the drug particle size was significantly reduced through Nano-formulation, resulting in a more uniform and monodisperse particle distribution.

### 9.2. Characterization of orientin and NF-Orientin using FT-IR

To assess the functional group similarities between the native and nano-formulated drugs, FT-IR analysis was conducted. The results visualised same number of peaks in the spectra for both the native Orientin and the NF-Orientin. In particular,

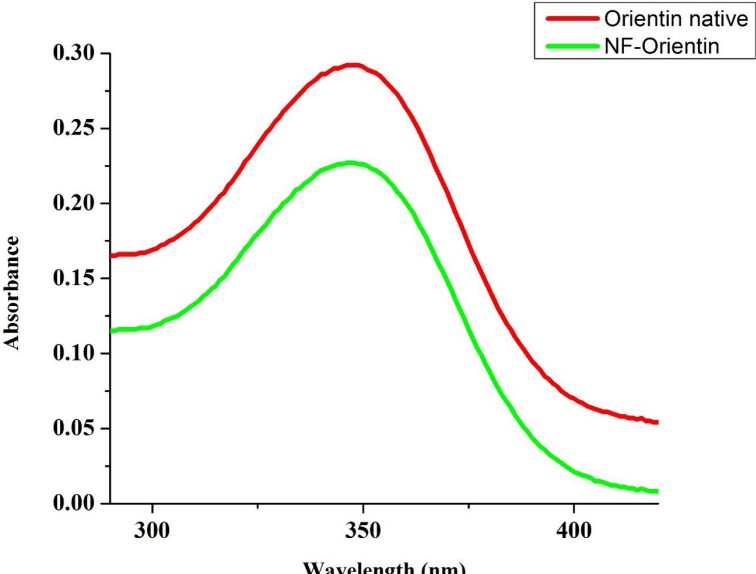

**Fig 5. UV-Spectrophotometer of Native Orientin and NF-Orientin.**

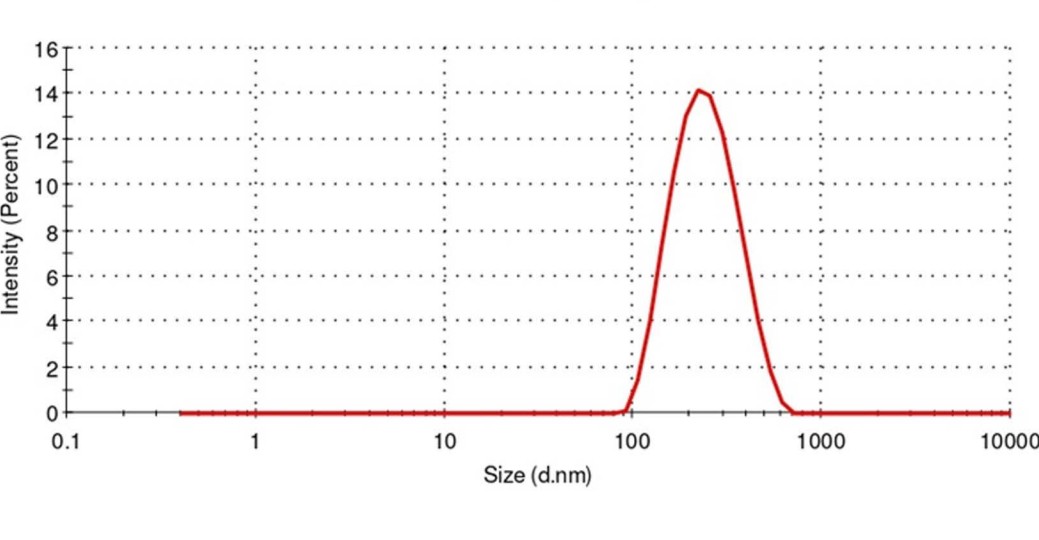

**Fig 6. Dynamic light scattering (DLS) data of NF-O.**

absorption peak at 3508 cm$^{-1}$ is linked to the accumulation of intramolecular hydrogen bonding and prominent hydroxyl O-H group is represented. Followed by a broad carboxylic acid peak at 3052 cm$^{-1}$ O-H stretch and a small sharp alkyne peak at 2316 cm$^{-1}$ confirming the C-H stretch. A medium sharp cyclic alkene peak at 1559 cm$^{-1}$ with a C=C stretching. Followed by a strong C-O stretching representing the alkyl aryl ether groups at 1258 cm$^{-1}$ and finally A strong peak at

795 cm-1 confirming the 1,2,3,4 – tetrasubstituted class representing the C-H bending. All these peaks are mention in the FT-IR image as shown in Fig 7. These findings demonstrate the presence of similar functional groups in both the two forms of native Orientin and the NF-Orientin, indicating that significant chemical properties were preserved throughout the Nano-formulation process.

### 9.3. NF-O inhibits the growth of MCF-7 cells

Using the MTT assay, the cytotoxicity effect for both native and NF-Orientin on the viability of MCF-7 cells was evaluated. The experimental results indicated significant changes in cell morphology and a substantial impact on cell viability following treatment with NF-Orientin compared to both untreated cells and those treated with native Orientin. At 10, 25, and 50μM, NF-Orientin reduced cell viability in a dose-dependent manner and the respective cell survival rates were 69%±2.0%, 51%±4.4%, and 37%±2.8%. On the other hand, native Orientin displayed cell viability rates of 78%±4.81%, 59%±5.1%, and 41%±3.5% at concentrations of 10, 25, and 50μM, respectively. NF-O treated cells exhibited significantly lower cell viability at all tested concentrations compared to cells treated with the corresponding concentrations of native Orientin. Notably, the vitality of cells treated with NF-Orientin was similar to that of doxorubicin (DOX), the positive control that is well-known for its lethal effects on cancerous cells. The IC50 values for both Native-O and NF-O, are 25.44 μg and 26.39 μg respectively. The MTT experiment results provide compelling evidence that NF-O demonstrates cytotoxic effects and significantly reduces cell viability compared to treatment with native Orientin alone. These findings suggest that NF-Orientin shows potential as an effective therapeutic approach for breast cancer cells, especially MCF-7 cells (Fig 8).

### 10. Wound healing assay

To examine the potential effects of native orientin and NF-O on cell behaviour, both compounds were administered at concentrations of 10 μM. The present experiment showed that NF-O had inhibitory effects on both cell growth and migration of the treatment group and showed a statistically significant increase compared to the control group. The results indicated that NF-O had a stronger impact on suppressing the migratory and proliferative properties of MCF-7 cells, in comparison to native orientin. These findings emphasize the potential of NF-O as a powerful modulator of cell behaviour and suggest its potential application in regulating the migratory and proliferative characteristics of cells (Figs 10–11). A significant reduction in wound thickness ($p < 0.01$) was observed with the application of 10 μM of native orientin and NF-orientin,

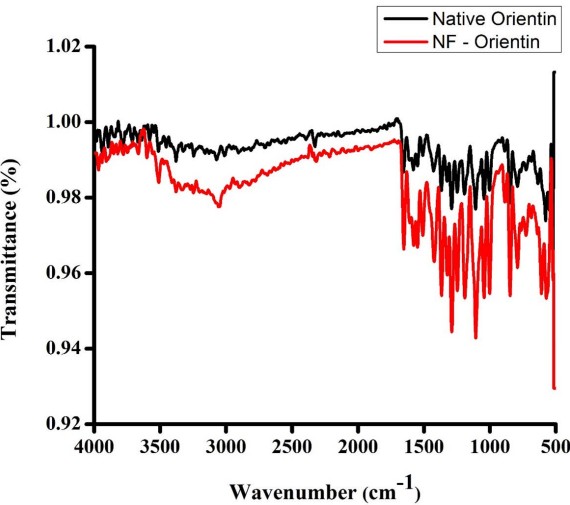

**Fig 7. Fourier Transform Infra-Red Spectroscopy (FT-IR) of Native Orientin and NF-Orientin.**

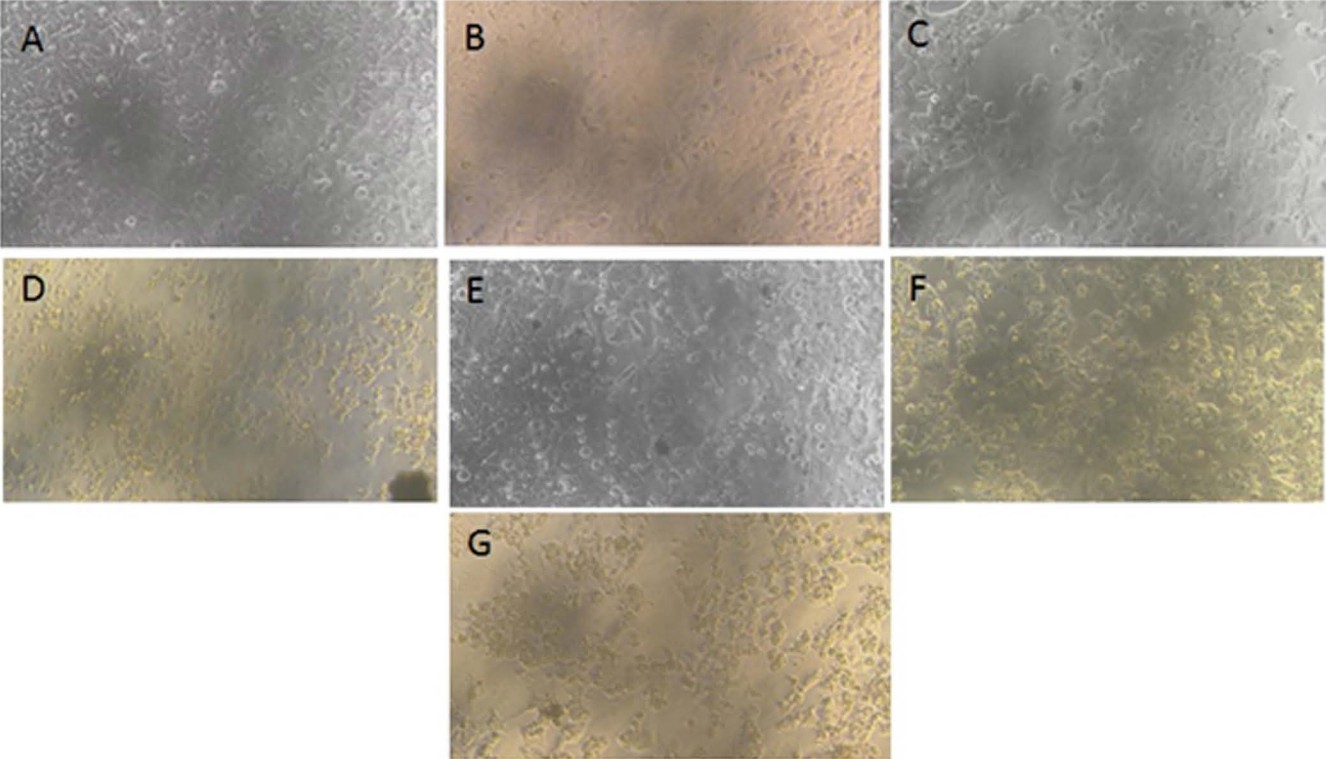

**Fig 8. Aggregation and morphology of MCF-7 cell lines A) Control, B) Native-O 10μM, C) Native-O 25μM, D) Native-O 50μM, E) NF-O 10μM, F) NF-O 25μM and G) NF-O 50μM depicts the effects of native as well as NF-O on the cell viability of breast cancer (MCF-7) cells.** Bright-field microscopy was utilized to examine the impact on cell morphology. The experimental groups included: (A) untreated MCF-7 cells, (B) 10μM of native orientin, (C) 30μM of native orientin, (D) 50μM of native orientin, (E) 10μM of NF-orientin, (F) 30μM of NF-orientin, and (G) 50μM of NF- orientin. Using MTT assay the viability of cells was assessed, with the data presented as the mean viability ± standard deviation from two independent experiments. Fig 9 shows the effects of different concentrations of native and NF-O on the viability of MCF7 cells remained illustrated via a graphical representation. In contrast to the native orientin treatment and the control group, the results showed that NF-O, at a concentration of 50 M, effectively suppressed the development of tumor cells (*P<0.05, **P<0.01, *** P<0.001*** P<0.0001).

resulting in approximately 45% wound closure at 24 hours, compared to around 85% closure in the control group. This finding highlights the strong potential of these formulations in inhibiting wound healing in a cell monolayer model.

## 11. NF-O inhibits neovascularization and blood vessels on CAM

*In ova*, the CAM membrane displayed evidence of angiogenesis. The eggs were treated to treatment with both native and nano-formulated orientin with a concentration of 10 µg/ml for five days. CAM morphometric analysis revealed that the NF-O drastically reduced density of blood vessels, amount of branch sites, total length of the vessel network, and overall number of nets as compared to the Native and control groups. Thin slices of tissue treated with hematoxylin and eosin to assess the blood vessel network (vasculature) and blood vessel formation (vasculogenesis) within the middle tissue layer (mesoderm) of the control chick embryo. Further, the control group had various blood vessel shapes in the middle layer. The membrane walls of the control CAM appeared normal and substantial. The CAM membrane, when treated with a concentration of 10 µg/ml of native orientin, displayed an abnormal morphology of blood vessels in the chorionic endoderm. In the mesodermal layer, a limited number of blood vessels and fibroblasts were noticed. In contrast, the CAM treated with a concentration of 10 µg/ml of NF-O exhibited the presence of fibroblasts in the connective tissue. The capillaries were

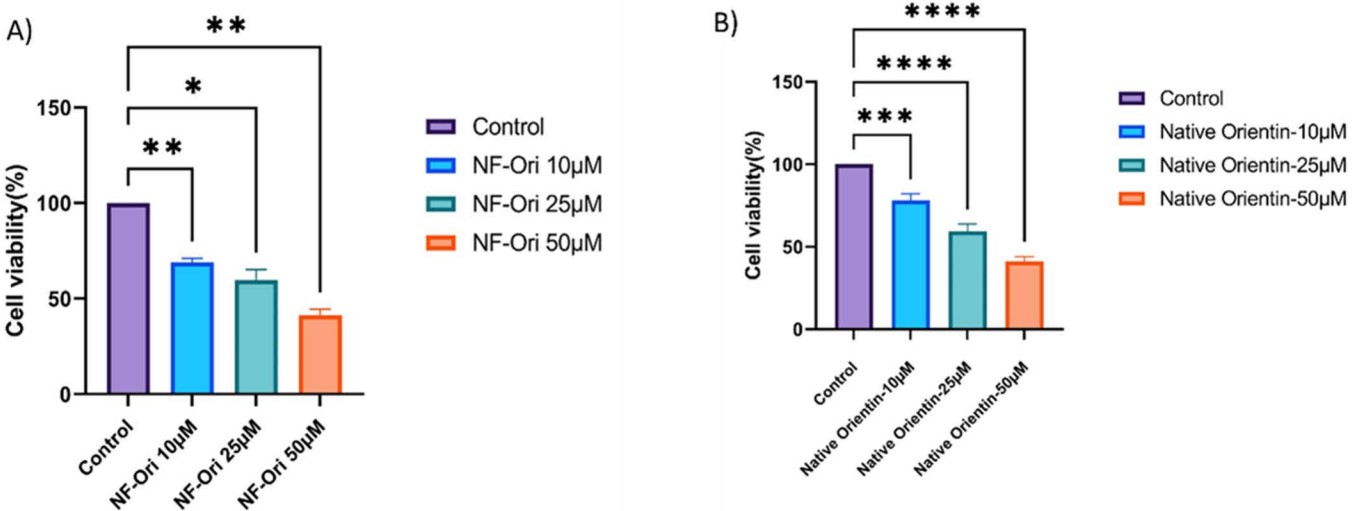

**Fig 9. Dose determination and cell viability estimation of (A) NF-Orientin and (B) Native Orientin using MCF-7 cell lines.**

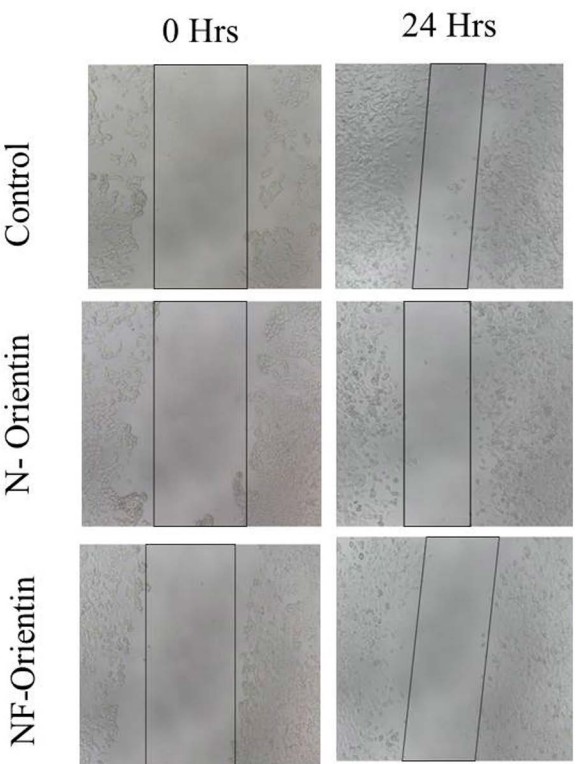

**Fig 10. Microscopic pictorial representation of wound healing assay (Control, native orientin (NF) and NF-O in 0 h and 24 h).**

**Wound Area Covered Overlay (MCF-7)**

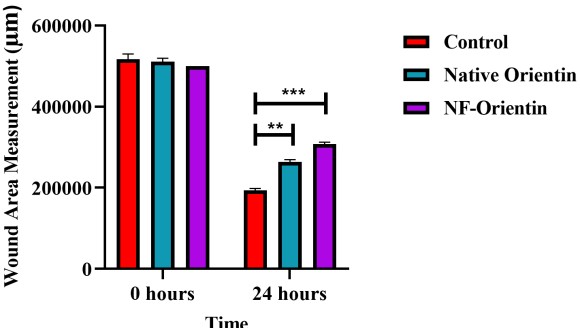

**Fig 11. Graphical representation of wound healing assay.**

reduced, and only a few small blood vessels with nucleated RBCs were observed in the mesodermal layer, in comparison to the native and control groups. These observations provide evidence that Nano-formulated Orientin alone demonstrated inhibition of blood vessels during angiogenesis as shown in Fig 12.

The impact of native-Orientin and nano-Orientin affect levels of messenger RNA (mRNA) for VEGF-A and FGF2 in the chorioallantoic membrane is shown in Fig 13. The researchers compared these gene expression levels between control groups, those treated with native-Orientin, and those treated with nano-Orientin. They used a statistical analysis called ANOVA to assess the significance of the results. The findings are presented as the average value ± standard deviation from three independent experiments. Asterisks indicate statistically significant differences compared to the control group (*$p < 0.05$, etc.,). In this study, the levels of VEGF-A and FGF2 mRNA expression in CAM tissue samples were examined using RT-PCR to assess the effects of the treatments. Results demonstrated a significant decrease in the expression of VEGF-A and FGF2 when compared with a control group and other similar compounds of native Orientin.

## 11.1. Histopathology examination of CAM

We assessed the morphological alterations of the CAM using histological sections stained with hematoxylin and eosin from the treated region to verify the anti-angiogenic properties of nano-formulated and native orientin. Blood arteries in the control CAM (which received no therapy) were of normal form, and observation revealed the presence of dense chorionic and allantoic epithelial layers. Examination of the stroma revealed the presence of abnormally large blood vessels. On the surrounds of the big blood arteries, proteinaceous fluid, and clogged minor blood capillaries were also observed. The CAM treated with Native Orientin had an abnormal description of blood vessels in the chorionic and allantoic epithelia. In comparison to the control, there were fewer big blood vessels visible in the stromal area. In addition, surrounding the major blood vessels were a few small blood capillaries. The CAM treated with NF-orientin displayed a relatively narrow chorionic and allantoic epithelial layer, as well as a decreased stromal area. The examination did not reveal any major blood vessels. Furthermore, a number of tiny blood vessels observed was much lower in comparison to the control group, and the presence of native orientin suggests suppression of sprouting as shown in Fig 14.

## 12. Discussion

Angiogenesis plays a crucial role in both physiological and pathological conditions, including cancer, diabetic retinopathy, and age-related macular degeneration [25]. Suppressing angiogenesis has been identified as a promising therapeutic approach to managing these diseases. This study aimed to assess the anti-angiogenic potential of nano-formulated Orientin (NF-O) compared to its natural form. The results indicate that NF-O exhibits enhanced anti-angiogenic and anti-tumor

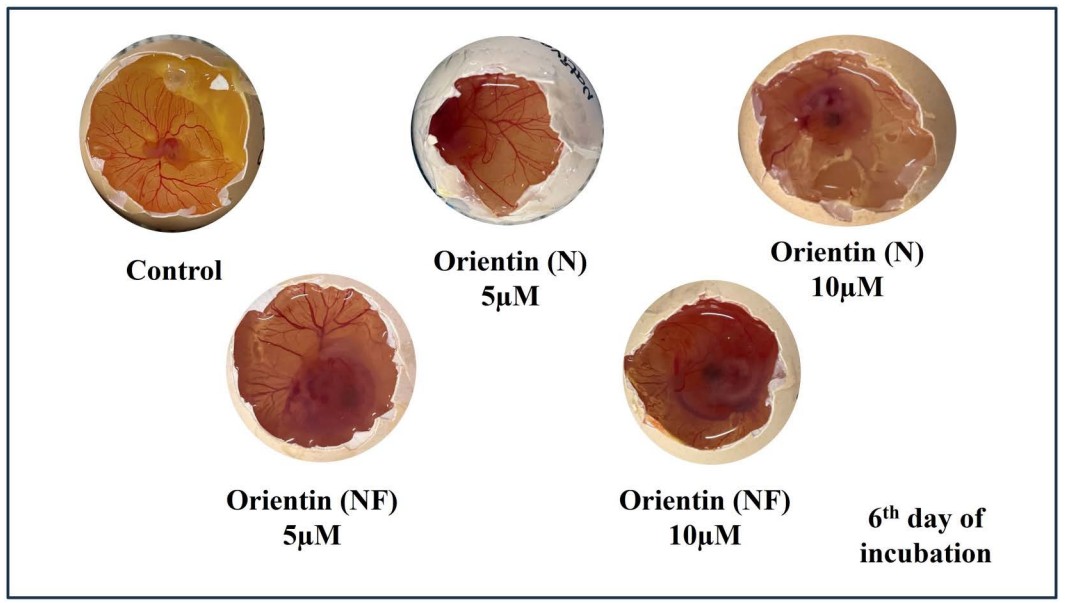

**Fig 12. Illustrates photographs of the CAM were captured to visualize the vascular plexus following the control group, native-orientin (5 μM), native-orientin (10 μM), NF-O (5 μM), and NF-O (10 μM).** 11.1. Gene Expression Analysis of Angiogenesis Over VEGF-A and FGF2.

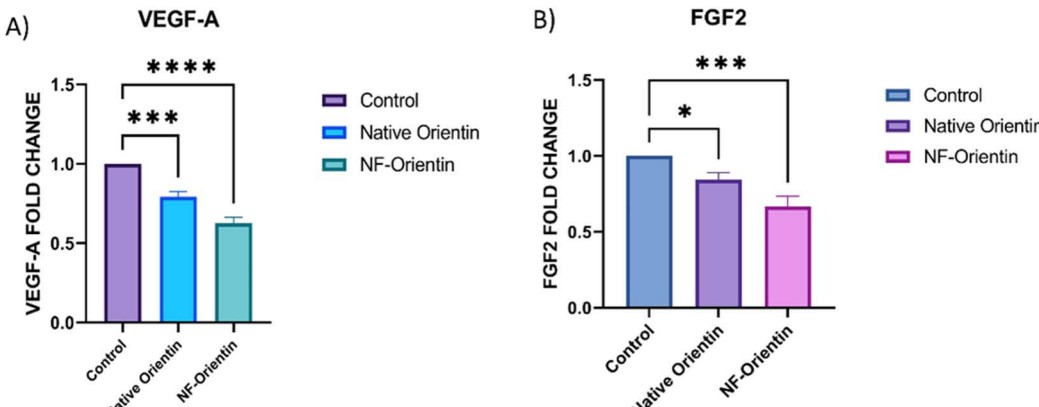

**Fig 13. Graphical representation on the impact of native orientin and NF-orientin on the expression of VEGF-A and FGF2 mRNA isolated from the CAM.**

effects, supporting its potential as a therapeutic candidate. Our findings demonstrated a significant reduction in cell viability upon NF-O treatment compared to native Orientin. This aligns with previous studies highlighting the advantages of nano-formulations in enhancing bioavailability and therapeutic efficacy [26]. The use of nano-carriers facilitates controlled drug release, improved cellular uptake, and protection against enzymatic degradation [27]. Furthermore, our investigation using the CAM model showed a marked reduction in vascular density and network complexity following NF-O treatment. These results are consistent with the work of Subbaraj et al. 2023 [4], who demonstrated the anti-angiogenic effects of kaempferol in combination with combretastatin, further strengthening the hypothesis that flavonoid-based nano-formulations can effectively inhibit aberrant vascularization. Nano-formulations such as liposomal and polymeric nanoparticles have

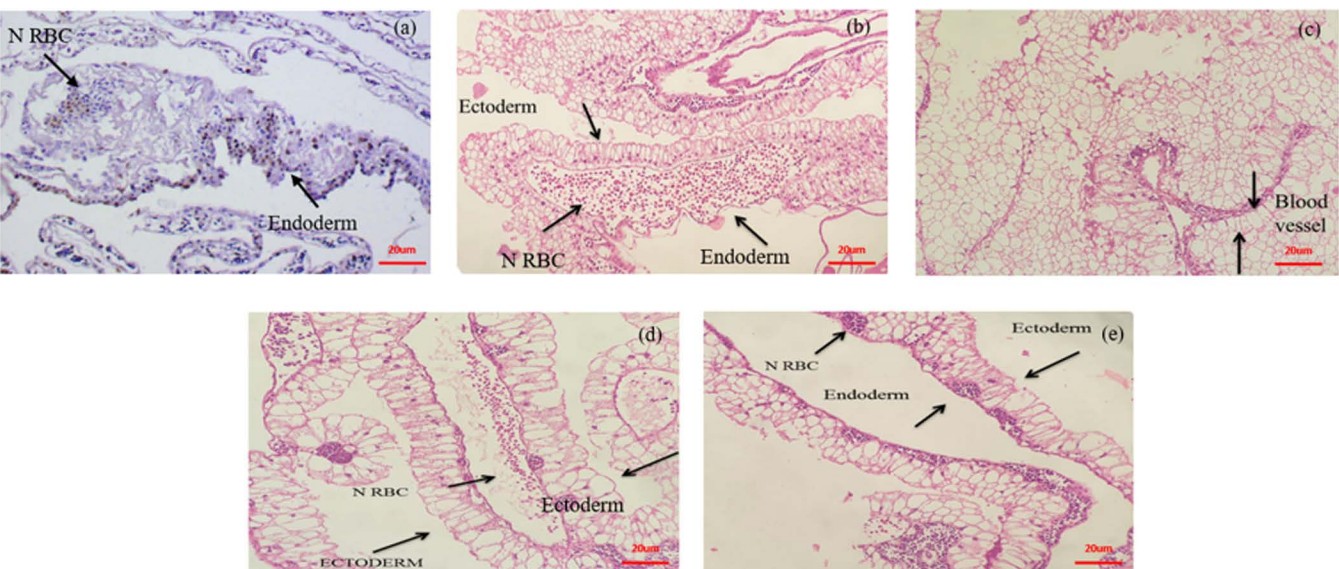

**Fig 14. H& E staining investigation of CAM treated with native-orientin (5 µM), native-orientin (10 µM), NF-O (5 µM), and NF-O (10 µM).**

been widely utilized to improve the pharmacokinetic profiles of bioactive compounds [28]. However, challenges such as limited drug loading capacity and potential toxicity of solvents used in nano-formulation must be carefully addressed [29]. In this study, solvent evaporation was employed as a preparation technique, offering advantages such as ease of synthesis and reproducibility. However, the toxicity and regulatory constraints of solvents such as DMSO and acetone remain significant concerns [30]. Future studies should explore alternative green synthesis methods or biocompatible carriers to optimize drug delivery and toxicity. The MTT assay revealed that NF-O exhibited superior cytotoxic effects on endothelial and cancer cells compared to native Orientin. This observation aligns with the research conducted [31], where nano-formulated flavonoids showed enhanced inhibition of cancer cell proliferation and migration. NF-O to show great inhibition of viability in MCF-7 human breast cancer cells with the influence that can be ascribed to the signaling pathways regarding growth and survival. That effect is interference with the cell signaling pathways involving cell growth and survival, just as predicted based on the results of our stable binding interactions detected through docking. Consistent with the intrinsic induction of apoptosis also seen in MCF-7 cells. Similarly, our wound healing assay indicated that NF-O significantly reduced cancer cell migration, further corroborating its therapeutic potential in metastatic cancer treatment. The activation of key molecular pathways, such as VEGF inhibition and modulation of nitric oxide signaling, may underpin these anti-angiogenic effects [32]. Further, wound healing assays corroborated the docking data as the NF-O treated samples showed significant inhibition of cell migration. The observation supports our docking data where predicted interactions were seen to contribute to the impairment of cellular mechanisms crucial for the motility and angiogenesis of cancer cells. Additionally, our DLS and FT-IR analyses confirmed that NF-O achieved a smaller, uniform particle size, which is crucial for its enhanced efficacy. A lower polydispersity index (PDI) ensures better stability and homogeneity in drug distribution, as also observed [33]. Studies suggest that nanoparticles around 220nm can greatly improve the dissolution of poorly soluble drugs. Various nanoparticle drug delivery systems, including liposomes, polymeric nanoparticles, and solid lipid nanoparticles, have been designed to enhance drug solubility [34]. These systems often employ particle sizes close to 220nm to optimize drug encapsulation, release, and absorption. Nanoscale particles can exist in different polymorphic forms, some of which may have higher solubility than their bulk counterparts. The 220nm size is particularly beneficial, as it is small enough to significantly enhance surface area while maintaining particle stability [35]. These physicochemical properties of NF-O contribute to its

superior biological activity by increasing cellular uptake and bioavailability [36]. The CAM membrane exhibited clear evidence of angiogenesis, with the control group showing normal blood vessel formation. However, treatment with both native Orientin and NF-O at a concentration of 10 µg/ml for five days resulted in significant morphological and molecular changes in blood vessel development. CAM morphometric analysis revealed that NF-O drastically reduced the density of blood vessels, the number of branch sites, the total length of the vessel network, and the overall number of vascular networks when compared to the native Orientin and control groups. Histological examination further confirmed these findings. In the control group, blood vessels within the middle tissue layer (mesoderm) exhibited normal morphology, with substantial and intact membrane walls. In contrast, treatment with native Orientin led to abnormal blood vessel structures in the chorionic endoderm, with fewer blood vessels and fibroblasts in the mesodermal layer. NF-O treatment resulted in a more pronounced anti-angiogenic effect, with a noticeable reduction in capillaries and only a few small blood vessels containing nucleated RBCs in the mesodermal layer. These observations strongly indicate that NF-O alone is effective in inhibiting blood vessel formation during angiogenesis. At the molecular level, our study assessed the impact of native Orientin and NF-O on the expression of key pro-angiogenic genes, VEGF-A and FGF2, in the CAM tissue. Using RT-PCR, we observed a significant decrease in VEGF-A and FGF2 mRNA expression levels in samples treated with NF-O compared to the control and native Orientin groups. These results suggest that NF-O effectively suppresses angiogenesis by downregulating critical genes involved in vascular development. Statistical analysis using ANOVA further confirmed the significance of these findings ($p < 0.05$). Histological analysis provided further evidence of the anti-angiogenic effects of NF-O. In the control CAM, blood vessels appeared normal, with dense chorionic and allantoic epithelial layers. The presence of abnormally large blood vessels and proteinaceous fluid around major blood arteries was also observed, indicating active angiogenesis. Treatment with native Orientin resulted in irregular blood vessel morphology in the chorionic and allantoic epithelia, along with a reduction in the number of large blood vessels in the stromal region. Additionally, small blood capillaries were observed around major blood vessels. Notably, NF-O treatment led to a significant reduction in the chorionic and allantoic epithelial layer thickness, as well as a marked decrease in the stromal area. The absence of major blood vessels and the significantly lower number of small blood capillaries in the NF-O group further support the strong anti-angiogenic properties of this formulation. Overall, this study underscores the promising therapeutic potential of NF-O as an anti-angiogenic and anti-cancer agent. Future research should focus on elucidating the precise molecular mechanisms underlying its enhanced efficacy, optimizing formulation strategies to further improve stability and drug loading, and conducting *in vivo* studies to validate its clinical applicability. The integration of NF-O into existing cancer treatment regimens holds significant promise for improving therapeutic outcomes and minimizing adverse effects.

## 13. Conclusion

The study's results underscore the ongoing success of *in vitro* and CAM prototypes in advancing research on assessing the anti-angiogenic properties. The *in vitro* experiment, which utilized NF-O, showcased its potential to inhibit cell proliferation while also being cost-effective, dependable, and time-efficient. Transforming the drug into a Nano-formulated version improved its delivery and binding capabilities, all the while preserving the same functional group and mechanisms. Analytical methods confirmed the equivalence of the Nano-formulated NF-O to its original form. Examination of cell viability and migration indicated that NF-O effectively hinders the growth of breast cancer cells. Furthermore, the CAM model provided additional evidence of the significant anti-angiogenic effects of the nano-formulated orientin when compared to its natural form. Subsequent research should delve deeper into unravelling the precise mechanism by which nano-formulated orientin impacts anti-angiogenesis and cancer progression.

## Supporting information

**S1 Data. MTT assay and Wound healing assay raw data.**
(XLSX)

## Acknowledgments

The authors would like to thank the management of Chettinad Academy of Research and Education (Deemed to be University) for providing facilities to perform this study.

## Author contributions

**Conceptualization:** K Nachammai.

**Data curation:** Kanu Shil.

**Formal analysis:** Kirubhanand Chandrasekaran, Langeswaran Kulanthaivel, Ram Kumar Anandan, Abdulhadi Ibrahim Bima, Zeenath Khan, Abdulhadi S. Burzangi, Noor A. Shaik.

**Investigation:** Yashwanth Elumalai, K Nachammai, Kirubhanand Chandrasekaran, Ram Kumar Anandan, Zeenath Khan, Abdulhadi S. Burzangi, Noor A. Shaik, Gowtham Kumar Subbaraj.

**Methodology:** Langeswaran Kulanthaivel, Sharon Benita Stephen.

**Project administration:** Sharon Benita Stephen.

**Supervision:** Langeswaran Kulanthaivel, Nuha Al-Rayes, Gowtham Kumar Subbaraj.

**Validation:** Abdulhadi Ibrahim Bima, Gowtham Kumar Subbaraj.

**Visualization:** Nuha Al-Rayes, Gowtham Kumar Subbaraj.

**Writing – original draft:** K Nachammai, Ram Kumar Anandan, Gowtham Kumar Subbaraj.

**Writing – review & editing:** K Nachammai, Langeswaran Kulanthaivel, Gowtham Kumar Subbaraj.

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
