## [Decision Letter · Decision Letter 0]

Dear Dr. Subbaraj,

Thank you for submitting your manuscript to PLOS ONE. After careful consideration, we feel that it has merit but does not fully meet PLOS ONE’s publication criteria as it currently stands. Therefore, we invite you to submit a revised version of the manuscript that addresses the points raised during the review process.

We look forward to receiving your revised manuscript.

Kind regards,

Chinnaperumal Kamaraj, Ph.D

Academic Editor

PLOS ONE

3. Thank you for stating the following financial disclosure:  [This research work was funded by Institutional Fund Projects under grant no. (IFPIP: 1174-290-1443). The authors gratefully acknowledge technical and financial support provided by the Ministry of Education and King Abdulaziz University, DSR, Jeddah, Saudi Arabia].  Please state what role the funders took in the study.  If the funders had no role, please state: "The funders had no role in study design, data collection and analysis, decision to publish, or preparation of the manuscript." If this statement is not correct you must amend it as needed.

5. In the online submission form, you indicated that your data will be submitted to a repository upon acceptance.  We strongly recommend all authors deposit their data before acceptance, as the process can be lengthy and hold up publication timelines. Please note that, though access restrictions are acceptable now, your entire minimal  dataset will need to be made freely accessible if your manuscript is accepted for publication. This policy applies to all data except where public deposition would breach compliance with the protocol approved by your research ethics board. If you are unable to adhere to our open data policy, please kindly revise your statement to explain your reasoning and we will seek the editor's input on an exemption.

Reviewers' comments:

Reviewer's Responses to Questions

**Comments to the Author**

1. Is the manuscript technically sound, and do the data support the conclusions?

Reviewer #1: Partly

Reviewer #2: Partly

2. Has the statistical analysis been performed appropriately and rigorously?

Reviewer #1: Yes

Reviewer #2: No

3. Have the authors made all data underlying the findings in their manuscript fully available?

Reviewer #1: Yes

Reviewer #2: Yes

4. Is the manuscript presented in an intelligible fashion and written in standard English?

Reviewer #1: Yes

Reviewer #2: Yes

Reviewer #1: This manuscript describes the Unleashing the anti-tumor angiogenic potential of nano-formulated orientin: in-silico, in-vitro, and in-ovo studies.

Abstract:

Reduce Introduction: Condense the introduction section, focusing on key points and omitting unnecessary details. Add results with relevant numerical data to strengthen the abstract.

Acronyms: Expand acronyms, such as "US," to "United States" upon their first usage.

Change the key word “In-vitro”

Statistical Data: Include statistical data specific to cancer incidence or prevalence in the author's country for relevance.

Flavonoids Introduction: Reduce the detailed discussion on “Flavonoids” to maintain focus.

Repetition: In the line, “Orientin, Orientin, formerly known...,” delete the repeated word "Orientin."

Study Objective: Elaborate on the novelty and objectives of the study at the end of the introduction.

Methods:

Chemical Quality: Rewrite the sentence “All other chemicals including solvents used in the present study were of high quality” as “All chemicals, including solvents used in the study, were of analytical or laboratory-grade quality, ensuring compliance with standard protocols.”

Cell Culture Methods: Elaborate on cell culture techniques with an emphasis on reproducibility and methodology details.

Formatting: Ensure uniform spacing between values and units, such as “10 mg” and “2 mL,” throughout the manuscript.

3D Structure and Docking: Provide detailed 3D structural representations with relevant amino acid residues, supported by dynamic simulation results in Figures 3 and 4.

Docking Validation: Validate molecular docking simulations with corresponding in vitro experimental data.

NF-O Size Concern: Address the concern regarding the large size of NF-O (220 nm) for nanoformulation in drug delivery and explain mitigation strategies.

Results:

Concentration Data: In the sentence, “The percentages of cell … at concentrations of 10 µM, 30 µM, and 50 µM, respectively,” delete repeated words for clarity.

IC50 Values: Include IC50 values in the experimental analysis.

Wound Healing: Write results from the wound healing assay more concisely.

FTIR Results: Make FTIR results more precise and validate data effectively.

Gene Expression: Conduct and report gene expression analysis with greater care to ensure accuracy.

The molecular mechanism through which the specific nanoparticles (NPs) induce biological activities must be investigated using additional data.

Discussion:

Improvement: Revise the discussion section extensively, incorporating insights from recent high-impact articles. This will enhance the depth and relevance of the analysis.

References:

Follow journal-specific referencing guidelines (e.g., “Gothai et al., 2017”).

Reviewer #2: Yeshwanth et al. reported results with the application of in silico, in vitro, and in ovo models to study the anti-angiogenic efficacy of a flavonoid, Orientin in nano-formulation. There are a few general and specific concerns.

Minor comments

In Fig. 9A, the X-axis is labeled as concentration. It can be labeled as 'test groups' or simply 'groups'. The legends are unnecessary in this figure since the group titles are already provided on the graph axis.

Add a reference scale to the histology images, i.e., magnification and size of structures or features within images.

The authors mentioned they have listed primer sequences in Table 1, but there are no such details.

The authors mentioned the docking score value of ligand-receptor complexes only in the text part of the results. The binding energy (Kcal/mol) values must also be included in respective columns in the table.

In PCR amplification protocol, mention of annealing and extension temperatures is missing in respective notes.

In the Q-PCR analysis, the authors did not provide sufficient statistical parameters applied. Further details on whether one-way or two-way ANOVA was used, and whether a secondary test / post-hoc analysis was performed, and if yes, which one? and justify if not applied.

Major comments

The control group is missing in the histopathology section to compare with the treated counterpart. Adding that would be appropriate which would provide a valid and logical reference to compare the changes and modifications observed after treatment.

In Figure 8, the microscopic images of the MTT assay are poor in terms of quality, clarity, and desired markings and focus. The concerned stress effects (aggregation, membrane disintegration/cell damage, cytotoxic effects) are not properly magnified. Authors need a kind of high-clarity/resolution images shot at 20X or 40X, whichever portrays convincingly a better visualization difference between control and treated cells.

While the present research is interesting with ample experiments, why are the angiogenic markers studied only in the CAM model, and not with a bit higher-scale setting – in vivo or ex vivo aortic ring assay or a similar kind? Why not authors in addition consider it to demonstrate their preliminary impression with more validation in the purview of anti-angiogenic potential. The chosen in-ovo model seems in this study not to be so adequate to support the effect of the tested compound in a practical angiogenic setting, hence, performing further in a higher setting would be a pivotal component of the present study, as well as for the scope and readers of the PLOS ONE.

**Do you want your identity to be public for this peer review?** For information about this choice, including consent withdrawal, please see our Privacy Policy

Reviewer #1: **Yes: ** Krishnan Raguvaran

Reviewer #2: **Yes: ** Guna Ravichandran

---

## [Author Response · Author response to Decision Letter 1]

10 Mar 2025

The Changes suggested by Reviewer 1 were highlighted in yellow colour.

The Changes suggested by Reviewer 2 were highlighted in green colour.

Reviewer #1: This manuscript describes the Unleashing the anti-tumor angiogenic potential of nano-formulated orientin: in-silico, in-vitro, and in-ovo studies.

1. Comment: Abstract and Reduce Introduction: Condense the introduction section, focusing on key points and omitting unnecessary details. Add results with relevant numerical data to strengthen the abstract.

Answer: Thank you for the valuable suggestion, as per the reviewer's comment, the introduction section is modified and made in a concise format and results were added in the abstract.

2. Comment: Acronyms: Expand acronyms, such as "US," to "United States" upon their first usage. Change the keyword “In-vitro”

Answer: Thank you for the valuable suggestion. As per the reviewer's comment, the acronyms were fully expanded, and the keywords were changed.

3. Comment: Statistical Data: Include statistical data specific to cancer incidence or prevalence in the author's country for relevance.

Answer: Thank you for the valuable suggestion. As per the reviewer's comments, the statistical data were included in the methodology section.

4. Comment: Flavonoids Introduction: Reduce the detailed discussion on “Flavonoids” to maintain focus.

Repetition: In the line, “Orientin, Orientin, formerly known...,” delete the repeated word "Orientin."

Answer: Thank you for the valuable suggestion. As per the reviewer's comment, I have reduced the detailed discussion on flavonoids as well as the repeated word orientin is removed.

5. Comment: Study Objective: Elaborate on the novelty and objectives of the study at the end of the introduction.

Answer: Thank you for the valuable suggestion. As per the reviewer's comments, the study objective is added and the novelty of the present study is elaborated in the introduction section. The following sentence was included in the revision of the manuscript; The novelty of the study includes the latest advancement of nanotechnology has the potential to transform drug delivery and greatly enhance the solubility of poorly soluble drugs. Due to their increased surface area and ability to penetrate biological barriers, nanoparticles are highly effective for drug delivery. These innovations could lead to the creation of more efficient and potent drug formulations for treating various diseases, including cancer. We aimed to assess the potential ability of native and nano-formulated orientin against cancer microenvironment. The parameters such as Molecular docking, Nano-characterization, Cell culture analysis, and the In-Ova CAM assay were performed.

6. Comment: Methods - Chemical Quality: Rewrite the sentence “All other chemicals including solvents used in the present study were of high quality” as “All chemicals, including solvents used in the study, were of analytical or laboratory-grade quality, ensuring compliance with standard protocols.”

Answer: Thank you for the valuable suggestions, as per the reviewer's comment, The revised sentence, "All chemicals, including solvents used in the study, were of analytical or laboratory-grade quality, ensuring compliance with standard protocols," clearly states the quality of the chemicals employed.

7. Comment: Cell Culture Methods- Elaborate on cell culture techniques with an emphasis on reproducibility and methodology details.

Answer: Thank you for the valuable suggestions. As per the reviewer's comment, the cell culture techniques were well emphasized and highlighted in the methodology section.

8. Comment: Formatting: Ensure uniform spacing between values and units, such as “10 mg” and “2 mL,” throughout the manuscript.

Answer: Thank you for the valuable suggestions. As per the reviewer's comment, I ensured uniform spacing between values and units.

9. Comment: 3D Structure and Docking: Provide detailed 3D structural representations with relevant amino acid residues, supported by dynamic simulation results in Figures 3 and 4.

Answer: Thank you for the valuable suggestions. As per the reviewer's comment, on the details of 3D structural representation and docking result. We include both 2D and 3D visualization of interaction details were updated in Figures 3 and 4. The figures can now represent every detail of interaction between the atoms of the respective molecules involved to focus on relevant amino acid residues.

10. Comment: Docking Validation: Validate molecular docking simulations with corresponding in vitro experimental data.

Answers: Thank you for the valuable suggestions. As per the reviewer's comment, our molecular docking analysis was intended for the study, which predicts the binding interactions and affinities between orientin and key targets for angiogenesis. From docking results, favorable interactions and key amino acid residues indicate that this can effectively inhibit pathways crucial for angiogenesis. To validate these predictions, we correlated our docking data with our in vitro experimental findings:

Anti-Angiogenic Activity

In silico docking simulations with angiogenesis-related targets EGFR (PDB ID: 2ITY), KRAS (PDB ID: 7LGI), NTRK (PDB ID: 7VKO), and ALK (PDB ID: 2XP2) show high affinity for molecular targets that regulate angiogenic signaling. This is validated by our in-ova CAM assay where the density of the vascular network was highly decreased on treatment with NF-O.

Anti-cancer activity

NF-O to show great inhibition of viability in MCF-7 human breast cancer cells with the influence that can be ascribed to the signaling pathways regarding growth and survival. That effect to interference with the cell signaling pathways involving cell growth and survival, just as predicted based on the results of our stable binding interactions detected through docking. Consistent with the intrinsic induction of apoptosis also seen in MCF-7 cells.

Inhibition of cellular migration

Further, wound healing assays corroborated the docking data as the NF-O treated samples showed significant inhibition of cell migration. The observation supports our docking data where predicted interactions were seen to contribute to the impairment of cellular mechanisms crucial for the motility and angiogenesis of cancer cells.

11. Comment: NF-O Size Concern: Address the concern regarding the large size of NF-O (220 nm) for nano-formulation in drug delivery and explain mitigation strategies.

Answer: Thank you for the valuable suggestions. As per the reviewer's comment, Research on drug nanoparticles consistently demonstrates that reducing particle size to the nanoscale significantly enhances dissolution rate and saturation solubility. Studies suggest that nanoparticles around 220nm can greatly improve the dissolution of poorly soluble drugs. Various Nanoparticle Drug Delivery Systems, including liposomes, polymeric nanoparticles, and solid lipid nanoparticles, have been designed to enhance drug solubility. These systems often employ particle sizes close to 220nm to optimize drug encapsulation, release, and absorption. The 220nm size range offers an ideal balance between increased surface area and particle stability, making it a key target for nano-formulation development. Nanoscale particles can exist in different polymorphic forms, some of which may have higher solubility than their bulk counterparts. The 220nm size is particularly beneficial, as it is small enough to significantly enhance surface area while maintaining particle stability. Extremely small nanoparticles can sometimes aggregate, reducing their effectiveness. Additionally, nanoparticle size influences their clearance from the bloodstream. Particles in the 220nm range may evade rapid clearance by the reticuloendothelial system (RES), extending circulation time and improving therapeutic efficacy. In our study, we observed that the NF-O had a potential effect on inhibiting anti-angiogenesis.

12. Comment: Results - Concentration Data: In the sentence, “The percentages of cell … at concentrations of 10 µM, 25 µM, and 50 µM, respectively,” delete repeated words for clarity.

Answer: Thank you for the valuable suggestions. As per the reviewer's comment, the repeated words and units were completely deleted from the manuscript.

13. Comment: IC50 Values: Include IC50 values in the experimental analysis.

Answer: Thank you for the valuable suggestions, as per the reviewer's comments we have included the IC50 values for both native orientin and NF-O, they are 25.44µg and 26.39µg respectively.

14. Comment: Wound Healing: Write results from the wound healing assay more concisely.

Answer: Thank you for the valuable suggestions. As per the reviewer's comment, the wound healing assay in the results section is concisely given for neat and better understanding.

15. Comment: FTIR Results: Make FTIR results more precise and validate data effectively.

Answer: Thank you for the valuable suggestions. As per the reviewer's comment, the FTIR results were added mentioning the functional groups of each peak that were obtained and given more precisely.

16. Comment: Gene Expression: Conduct and report gene expression analysis with greater care to ensure accuracy.

Answer: Thank you for the valuable suggestions. As per the reviewer's comment, the gene expression analysis was performed for the in-ova CAM model for two genes VEGF-A and FGF2 and the results demonstrated that a significant decrease in the expression of VEGF-A and FGF2 when compared with a control group treated with native orientin and NF-O.

17. Comment: The molecular mechanism through which the specific nanoparticles (NPs) induce biological activities must be investigated using additional data.

Answer: Thank you for the valuable suggestions. As per the reviewer's comment, the need to elucidate the molecular mechanisms underlying the observed biological activities of the nanoparticles (NPs). We acknowledge that the current study provides preliminary data on the effects of NF-Orientin by solvent evaporation method and, as such, focuses primarily on establishing the correlation between Native-Orientin and NF-Orientin exposure and the observed biological responses. A detailed investigation of the specific molecular mechanisms involved is indeed crucial for a deeper understanding of the NF-orientin interactions with biological systems. Due to the scope and preliminary nature of this work, a comprehensive mechanistic investigation was beyond the capacity of the present study. However, we fully agree with the reviewer on the importance of this aspect and are currently planning future studies specifically designed to address these mechanistic questions in a more expanded and targeted manner. We aim to believe future work will build upon the foundation laid by this study and provide a more complete picture of the biological impact of these NF-Orientin.

18. Comment: Discussion Improvement: Revise the discussion section extensively, incorporating insights from recent high-impact articles. This will enhance the depth and relevance of the analysis.

Answer: Thank you for the valuable suggestions. As per the reviewer's comment, the discussion part was revised by incorporating high-impact journal articles that were included in the manuscript.

19. Comment: References:

Follow journal-specific referencing guidelines (e.g., “Gothai et al., 2017”).

Answer: Thank you for the valuable suggestions. As per the reviewer's comment, the following above-mentioned reference is removed from the manuscript.

Reviewer #2: Yeshwanth et al. reported results with the application of in silico, in vitro, and in-ovo models to study the anti-angiogenic efficacy of a flavonoid, Orientin in nano-formulation. There are a few general and specific concerns.

Minor comments

1. Comment: In Fig. 9A, the X-axis is labeled as concentration. It can be labeled as 'test groups' or simply 'groups'. The legends are unnecessary in this figure since the group titles are already provided on the graph axis.

Answer: Thank you for the valuable suggestions. As per the reviewer's comment, in Figure 9A, the X-axis which was labeled as concentrations is now labeled with groups.

2. Comment: Add a reference scale to the histology images, i.e., magnification and size of structures or features within images.

Answer: Thank you for the valuable suggestions. As per the reviewer's comment, the reference scale to the histology images was added within the images.

3. Comment: The authors mentioned they have listed primer sequences in Table 1, but there are no such details.

Answer: Thank you for the valuable suggestions. As per the reviewer's comment, the primer sequence which was earlier given is now modified and mentioned in Table 1.

4. Comment: The authors mentioned the docking score value of ligand-receptor complexes only in the text part of the results. The binding energy (Kcal/mol) values must also be included in respective columns in the table.

Answer: Thank you for the valuable suggestions. As per the reviewer's comment, the updated version of Table 2 with the binding energy (Kcal/mol) values was included.

5. Comment: In PCR amplification protocol, mention of annealing and extension temperatures is missing in respective notes.

Answer: Thank you for the valuable suggestions. As per the reviewer's comment, we have included the PCR amplification protocol mentioning the Initial denaturation at 90° C for 30 s, annealing at 60°C for 30s, and extension at 72°C for 30s for 40 cycles were programmed during amplification.

6. Comment: In the Q-PCR analysis, the authors did not provide sufficient statistical parameters applied. Further details on whether one-way or two-way ANOVA was used, whether a secondary test / post-hoc analysis was performed, and if yes, which one? and justify if not applied.

Answer: Thank you for the valuable suggestions. As per the reviewer's comment, we have included all the statistical parameters in detail which were used for evaluating the significance in the methodology section. The results are presented as mean ± SD based on the number of experiments conducted. Statistical analysis was performed using one-way ANOVA in GraphPad Prism 8. Following a significant ANOVA result, the Newman-Keuls post-hoc test was applied to compare group means. A significance threshold of p<0.05 was set to determine statistically significant differences between groups.

Major comments

7. Comment: The control group is missing in the histopathology section to compare with the treated counterpart. Adding that would be appropriate which would provide a valid and logical reference to compare the changes and modifications observed after treatment.

Answer: Thank you for the valuable suggestions. As per the reviewer's comment, the control group images of histopathological analysis were included to compare with the treated counterparts.

8. Comment: In Figure 8, the microscopic images of the MTT assay are poor in terms of quality, clarity, and desired markings and focus. The concerned stress effects (aggregation, membrane disintegration/cell damage, cytotoxic effects) are not properly magnified. Authors need a kind of high-clarity/resolution images shot at 20X or 40X, whichever portrays convincingly a better visualization difference between control and treated cells.

Answer: Thank you for the valuable suggestions. As per the reviewer's comment, In Figure 8, the microscopic images that we have given for the MTT assay were captured using a high-resolution microscope. We understand your concern regarding the high resolution of images, but, so far as the images included in the results are to the best of our knowledge and potential. We kindly request you to accept the images for further proceedings. In future studies, we will take the necessary implications to provide even better-quality images.

9. Comment: While the present research is interesting with ample experiments, why are the angiogenic markers studied only in the CAM model, and not with a bit higher-scale setting – in vivo or ex vivo aortic ring assay or a similar kind? Why not authors in addition consider it to demonstrate their preliminary impression with more validation in the purview of anti-angiogenic potential

---

## [Decision Letter · Decision Letter 1]

Unleashing the Anti-tumor Angiogenic Potential of Nano-Formulated Orientin: In Silico, In Vitro, and In Ovo Studies

PONE-D-24-48356R1

Dear Dr. Subbaraj,

We’re pleased to inform you that your manuscript has been judged scientifically suitable for publication and will be formally accepted for publication once it meets all outstanding technical requirements.

Kind regards,

Chinnaperumal Kamaraj, Ph.D

Academic Editor

PLOS ONE

Additional Editor Comments (optional):

We're delighted to let you know that your manuscript has been accepted for publication in PLOS One.

Reviewers' comments:

Reviewer's Responses to Questions

**Comments to the Author**

Reviewer #1: All comments have been addressed

Reviewer #2: All comments have been addressed

Reviewer #3: All comments have been addressed

2. Is the manuscript technically sound, and do the data support the conclusions?

Reviewer #1: Yes

Reviewer #2: Yes

Reviewer #3: Yes

3. Has the statistical analysis been performed appropriately and rigorously?

Reviewer #1: Yes

Reviewer #2: Yes

Reviewer #3: Yes

4. Have the authors made all data underlying the findings in their manuscript fully available?

Reviewer #1: Yes

Reviewer #2: Yes

Reviewer #3: Yes

5. Is the manuscript presented in an intelligible fashion and written in standard English?

Reviewer #1: Yes

Reviewer #2: Yes

Reviewer #3: Yes

Reviewer #1: (No Response)

Reviewer #2: The authors have satisfactorily addressed the minor comments. And hope they will implicate other revisions in future experiments as they would validate the findings for the desired efficacy.

Reviewer #3: The author did excellent work for anticancer and in silico work. The author addresses all the comments. Now can proceed to publication in the PLOS Journal

**Do you want your identity to be public for this peer review?** For information about this choice, including consent withdrawal, please see our Privacy Policy

Reviewer #1: **Yes: ** Krishnan Raguvaran

Reviewer #2: **Yes: ** Dr. Guna Ravichandran

Reviewer #3: No

---

## [Editor Report · Acceptance letter]

PONE-D-24-48356R1

PLOS ONE

Dear Dr. Subbaraj,

I'm pleased to inform you that your manuscript has been deemed suitable for publication in PLOS ONE. Congratulations! Your manuscript is now being handed over to our production team.

Kind regards,

on behalf of

Dr. Chinnaperumal Kamaraj

Academic Editor

PLOS ONE